# Sequencing and Comparative Analysis of the Chloroplast Genome of *Angelica polymorpha* and the Development of a Novel Indel Marker for Species Identification

**DOI:** 10.3390/molecules24061038

**Published:** 2019-03-15

**Authors:** Inkyu Park, Sungyu Yang, Wook Jin Kim, Jun-Ho Song, Hyun-Sook Lee, Hyun Oh Lee, Jung-Hyun Lee, Sang-Nag Ahn, Byeong Cheol Moon

**Affiliations:** 1Herbal Medicine Resources Research Center, Korea Institute of Oriental Medicine, Naju 58245, Korea; pik6885@gmail.com (I.P.); sgyang81@kiom.re.kr (S.Y.); ukgene@kiom.re.kr (W.J.K.); songjh@kiom.re.kr (J.-H.S.); 2Department of Agronomy, College of Agriculture and Life Sciences, Chungnam National University, Daejeon 34134, Korea; leehs0107@gmail.com (H.-S.L.); ahnsn@cnu.ac.kr (S.-N.A.); 3Phyzen Genomics Institute, Seongnam 13558, Korea; dlgusdh88@phyzen.com; 4Department of Biology Education, Chonnam National University, Gwangju 77, Korea; quercus@jnu.ac.kr

**Keywords:** *Angelica polymorpha*, *Ligusticum officinale*, plastid, herbal medicine, molecular marker

## Abstract

The genus *Angelica* (Apiaceae) comprises valuable herbal medicines. In this study, we determined the complete chloroplast (CP) genome sequence of *A. polymorpha* and compared it with that of *Ligusticum officinale* (GenBank accession no. NC039760). The CP genomes of *A. polymorpha* and *L. officinale* were 148,430 and 147,127 bp in length, respectively, with 37.6% GC content. Both CP genomes harbored 113 unique functional genes, including 79 protein-coding, four rRNA, and 30 tRNA genes. Comparative analysis of the two CP genomes revealed conserved genome structure, gene content, and gene order. However, highly variable regions, sufficient to distinguish between *A. polymorpha* and *L. officinale*, were identified in hypothetical chloroplast open reading frame1 (*ycf1*) and *ycf2* genic regions. Nucleotide diversity (*Pi*) analysis indicated that *ycf4*–chloroplast envelope membrane protein (*cemA*) intergenic region was highly variable between the two species. Phylogenetic analysis revealed that *A. polymorpha* and *L. officinale* were well clustered at family Apiaceae. The *ycf4*-*cemA* intergenic region in *A. polymorpha* carried a 418 bp deletion compared with *L. officinale*. This region was used for the development of a novel indel marker, LYCE, which successfully discriminated between *A. polymorpha* and *L. officinale* accessions. Our results provide important taxonomic and phylogenetic information on herbal medicines and facilitate their authentication using the indel marker.

## 1. Introduction

In plants, the chloroplast (CP) plays an important role in photosynthesis and carbon fixation as well as starch, fatty acid, and amino acid biosynthesis [1]. In higher plants, the CP genome ranges in size from 120 to 180 Kb and has a quadripartite structure, including a large single copy (LSC) region, a small single copy (SSC) region, and two copies of an inverted repeat (IR) region (IRa and IRb) [2]. Angiosperm CP genomes usually contain 110–130 genes, with up to 80 protein-coding genes, approximately 30 transfer RNA (tRNA) genes, and four ribosomal RNA (rRNA) genes [3]. Despite the highly conserved genome structure, gene content, and gene order among plant species, CP genomes exhibit genomic arrangement, IR region loss, and gene loss in some angiosperms such as parasitic plants [4,5,6,7,8,9]. Advances in next generation sequencing technologies have reduced the cost and complexity of CP genome assembly compared with Sanger sequencing [10]. The CP genome has been widely used for phylogenetic analysis and molecular marker development in plant species. These molecular markers are highly useful DNA barcoding tools for the authentication and identification of plant taxa including herbal medicines. For example, *matK* and *rbcL* genes in CP genomes are used as universal plant DNA barcodes [11]. The genus *Angelica* comprises valuable herbal medicines [12]. Although the CP genomes of a few *Angelica* species have been reported [13,14,15,16,17] and are available from GenBank, limited genomic information is available for the identification of *Angelica* species. Additional genomic information is needed to understand the utility of herbal medicines in the genus *Angelica*.

Insertions/deletions (indels) in CP genomes occur because of genomic rearrangements resulting from slipped strand mispairing, stem-loop secondary structure, and intramolecular recombination [18,19,20]. Indels represent intraspecific polymorphisms in plant populations [21] and are used for species identification. Indel in the *trnL-F* region has been widely used as a universal DNA barcode for species classification [22]. Phylogenetic relationships among 41 *Poa* species have been determined using indels in *trnL-F* and *trnL* introns, clustering these species into four major groups [23]. Indels in *trnL-F*, *trnG-trnS*, and *trnL* introns have been used for the analysis of the CP genomes of *Silene latifolia* and *S. vulgaris*. Furthermore, the authors showed that indels evolved at slightly higher rates than single nucleotide polymorphisms (SNPs) in the *Silene* genus [24]. In the genus *Aconitum*, four species have been distinguished using indels in CP genomes [25,26]. Furthermore, CP genome indels have been used to identify intraspecific variation in the genera *Fagopyrum* (*F. tataicum* vs. *F. esculentum*) and *Ipomoea* (*I. nil* vs. *I. purpurea*) [27,28]. Thus, indels in CP genomes are useful for phylogenetic and evolutionary analyses of plant species as well as for species identification.

The genus *Angelica* (family Apiaceae) is a taxonomically complex and controversial group comprising approximately 110 species with diverse morphology [29]. *Ligusticum officinale* is widely distributed in East Asia, and dried rhizomes of this species are used as an important herbal medicine [12]. Unfortunately, *A. polymorpha* is frequently misused as *L. officinale* in inauthentic preparations of herbal medicines in Korean herbal markets because sliced preparations of the two species are highly similar to the naked eye. To ensure a consistent pharmacological effect of herbal medicines, accurate identification of *L. officinale* and *A. polymorpha* is essential. Therefore, an objective method of analysis, such as a molecular marker, is needed for the identification of herbal medicines.

In this study, we characterized the CP genome of *A. polymorpha* and compared it with that of *L. officinale* with the aim to identify highly variable regions and understand the phylogenetic relationship between the two species. Additionally, we aimed at developing an efficient molecular marker to distinguish between the CP genomes of these species.

## 2. Results and Discussion

### 2.1. CP Genome Organization of A. polymorpha

The CP genome of *A. polymorpha* was sequenced using the Illumina MiSeq platform. Sequencing at approximately 75× coverage generated 1.25 Gb of paired-end reads (Appendix A). The complete circular CP genome of *A. polymorpha*, completed after gap filling and manual editing, was 147,121 bp in length. Paired-end read mapping was conducted to validate the draft genome (Appendix A). The CP genome of *A. polymorpha* showed a quadripartite structure like in most land plants consisting of a pair of IR regions (17,870 bp each) separated by LSC (93,591 bp) and SSC (17,796 bp) regions (Figure 1, Table 1). The CP genome of *A. polymorpha* was AT-rich (62.5%), and the AT content of LSC (64.1%) and SSC (69%) regions was higher than that of the IR regions (54.9%); these data are consistent with those of other angiosperm CP genomes [2,30]. Sequences of the junctions between the LSC, SSC, and IR regions were validated using PCR-based sequencing (Appendix A). The CP genome of *A. polymorpha* harbored 113 predicted genes, of which 97 were present as single copies in the LSC and SSC regions, while 17 were duplicated in the IR regions (Appendix A). The 97 unique genes included 79 protein-coding genes, 30 tRNA genes, and four rRNA genes. Additionally, the CP genome of *A. polymorpha* harbored 17 intron-containing genes. Among these, 14 genes (nine protein-coding and five tRNA genes) contained a single intron, while two genes (*ycf3* and *clpP*) contained two introns (Appendix A). Of the 17 intron-containing genes, 12 genes (nine protein-coding and three tRNA genes) were located in the LSC region, one protein-coding gene in the SSC region, and four genes (two protein-coding and two tRNA genes) in the IR regions. Of the 79 protein-coding genes, six genes (*ndhB*, *rpl2*, *rpl23*, *rps7*, *rps12* and *ycf15*) were duplicated in the IR regions. The start codons of *ndhD* and *rps19* were ACG and GTG, respectively, which were used as an alternative to ATG. The use of ACG and GTG as start codons is a common phenomenon in various genes in CP genomes of land plants [31,32,33]. The protein-coding genes comprised 21,587 bp in the CP genome of *A. polymorpha* (Appendix A), and codons of leucine and isoleucine were highly abundant (Appendix A). Relative synonymous codon usage (RSCU) values revealed synonymous codon usage bias, with a high proportion of synonymous codons harboring A or T(U) nucleotide in the third position (Appendix A). Overall, the genome structure, gene number, and codon usage in the CP genome of *A. polymorpha* were consistent with those in CP genomes of other *Angelica* species [13,15,17].

### 2.2. Analysis of Repeated Sequences in the CP Genomes of A. polymorpha and L. officinale

Repeated sequences were abundant in the CP genomes of both species. These repeat sequences result in structural variation due to genomic rearrangement, gene expansion, and pseudogene formation [8,35]. Simple sequence repeats (SSRs), also known as microsatellites, comprise 1–6 nucleotides [36]. We analyzed SSRs in the CP genomes of the two species (Appendix A). The CP genomes of *A. polymorpha* and *L. officinale* harbored a similar number of SSRs (209 and 203, respectively). Most of these SSRs were located in single copy regions (LSC and SSC), as expected. The number of SSRs was similar between the SSC and IR regions. SSRs were more abundant in the intergenic spacer (IGS) region, especially the non-coding region, than in genic regions, and mononucleotide motifs were the most abundant type of repeats, followed by dinucleotide motifs, in both CP genomes (Appendix A). We also identified tandem repeats (>20 bp) in the two CP genomes (Appendix A). Most of the tandem repeats (20–59 bp) were located in IGS and LSC regions. The longest tandem repeat (100 bp) was present in the CP genome of *L. officinale*. Palindromic repeats were located in the LSC region in both CP genomes (Appendix A). Overall, the CP genomes of *A. polymorpha* and *L. officinale* showed a similar number and type of repeats, and no polymorphism was detected between the two genomes.

### 2.3. Comparative Analysis of the CP Genomes of A. polymorpha and L. officinale

The IR regions represent the most highly conserved sequences in the CP genome [37]. The contraction and expansion of sequences at the borders of IR regions is a common evolutionary event, which is mainly responsible for variation in CP genome size and genomic rearrangement [38]. In this study, we analyzed the border structure of LSC, SSC, and IR regions in the two CP genomes (Figure 2). The *ycf2* gene was located at the LSC/IRa junction. The *ycf1* pseudogene and *ycf1* gene, which was located at the IRa/SSC and SSC/IRb junctions, extended into the SSC region. The location of most other genes was similar to their location in other CP genomes [28,39].

Gene content, order, and orientation were similar between the CP genomes of *A. polymorpha* and *L. officinale*. To compare CP genomes of the two species, we performed multiple sequence alignment of the whole CP genome sequences using mVISTA (Figure 3). The non-coding region was more variable than the coding region, and the IGS region was the most variable in both CP genomes. Five highly variable regions were identified in this study including three IGS regions (*trnT-psbD*, *ycf4-cemA*, and *ycf2-trnL*) and two genic regions (*ycf1* and *ycf2*). To determine sequence divergence between the CP genomes of *A. polymorpha* and *L. officinale*, we calculated the nucleotide diversity (Pi) of the CP genome sequences (Figure 4). The IR regions were more highly conserved than the LSC and SSC regions, with average Pi values of 0.002 in IR regions and 0.009 in single copy regions (with some IR regions showing a *Pi* value of 0). In the LSC, *ycf4-cemA* exhibited a *Pi* of 0.189, which was the highest. Although the CP genomes of both species were mostly highly conserved, the IGS regions showed divergence. High divergence in the IGS regions, including *trnT-psbD*, *ycf4-cemA*, and *ycf2-trnL*, because of the presence of indels, SNPs, and structural variation has been previously reported in CP genomes of other plant species [40,41,42,43]. In this study, the *ycf4-cemA* region was used for the development of a molecular marker to distinguish between *A. polymorpha* and *L. officinale*.

### 2.4. Phylogenetic Relationship between A. polymorpha and L. officinale

The CP genomes are valuable genomic resources for the reconstruction of accurate high-resolution phylogenies [44,45]. To identify the phylogenetic positions of *A. polymorpha* and *L. officinale* within the Apiaceae family, 52 protein-coding sequences shared by 33 CP genomes were aligned over a total length of 38,279 bp (Figure 5). The maximum likelihood (ML) tree and Bayesian inference (BI) trees contained 22 of 30 nodes, with ML bootstrap values of 100% and BI posterior probabilities of 1.0. Both the ML and BI phylogenetic results indicated that Apiaceae and Araliaceae with ML bootstrap values of 100% and BI posterior probabilities of 1.0. *L. tenuissimum* and *L. officinale* clustered together. Moreover, these two *Ligusticum* species were closely related to *Coriandrum sativum* within Apiaceae. *A. polymorpha* was well-positioned within the genus *Angelica*. *Foeniculum vulgare* and *Anethum graveolens* formed a monophyletic group and a sister relationship with *Petroselinum crispum* within Apiaceae. The genus *Angelica* showed highly ML bootstrap values and BI posterior probabilities, species within this genus were well clustered according to the APG IV system [46]. However, *Glehnia littoralis* weakly clustered within the genus *Angelica* in this study. In a previous study, phylogenetic trees of Apiaceae were reconstructed using internal transcribed spacer (ITS) and CP loci [29,47,48], and our results were consistent with phylogenetic trees based on both ITS and CP loci. Genera *Glehnia* and *Angelica* showed different morphological characteristics, and their phylogenetic relationship was not clear based on whole CP genome sequences. However, to understand the phylogenetic relationship between *Angelica* and *Glehnia* species, in-depth investigation of other CP genomes and reinterpretation of morphological data are needed. Furthermore, taxonomic delimitation of the following four species at the genus level has changed depending on the view point of taxonomists [49,50,51,52]: *Ledebouruella seseloides* (=*Saposhnikovia divaricata* (Turcz.) Schischk), *L. tenuissimum* (=*Conioselinum tenuissimum* (Nakai) Pimenov & Kljuykov), *L. officinale* (=*Cnidium officinale* Makino), and *Peucedanum insolens* [=*Sillaphyton podagraria* (H. Boissieu) Pimenov]. Among these species, *L. tenuissimum* and *L. officinale* clustered within a monophyletic group in this study. We suggest that the *Ligusticum* taxa should be considered for further investigation. Taken together, our results provide insights into the phylogenetic relationship among species within Apiaceae.

### 2.5. Development and Validation of an Indel Marker for Authentication of Cnidii Rhizoma

In this study, we identified divergent regions in the CP genomes of *A. polymorpha* and *L. officinale* to distinguish between these two species. Results showed that the CP genome of *A. polymorpha* carries a 418 bp deletion in the *ycf4-cemA* region compared with *L. officinale*. To characterize these sequences, we aligned these sequences with those available in the non-redundant (NR) database of NCBI. Multiple sequence alignment revealed species-specific sequences but no copy number variation of tandem repeats. To develop indel markers, sequence-specific primers were designed in the conserved regions flanking *ycf4* and *cemA* (Table 2). The LYCE primers successfully amplified sequences from both *L. officinale* and *A. polymorpha* (Figure 6). The indel marker was tested on 21 accessions collected from different sites in Korea using LYCE primers. These 21 samples were clearly distinguished into 12 *L. officinale* and nine *A. polymorpha* samples (Appendix A). The CP DNA fragments amplified from the tested samples were sequenced to determine the exact amplicon size. The LYCE primer pair amplified a 540 bp amplicon from *L. officinale* samples and a 122 bp fragment from *A. polymorpha* samples. The predicted sizes of insertions or deletions in the CP genomes were consistent with fragment sizes amplified from *L. officinale* and *A. polymorpha* samples.

Dried rhizomes of *L. officinale* are used as a traditional herbal medicine in Korea [12]. Although phylogenetic analysis indicated that *A. polymorpha* is distant from *L. officinale*, a molecular approach is needed for efficient differentiation between authentic herbal medicines and adulterants that appear similar because of similar shaped rhizomes and sliced herbal products. Indels in CP genomes were useful for species identification and distinguishing between authentic and inauthentic herbal medicines. Previous studies have reported indel markers of CP genomes [26,53,54]. *Aconitum pseudolaeve*, *A. longecassidatum*, and *A. barbatum* have been clearly distinguished on the basis of variation in CP genomes using indel markers [25]. Similarly, species identification of *F. tataricum* and *F. esculentum* has been performed using the same approach [27]. Thus, indel markers play an important role in species identification and herbal medicine authentication. The LYCE indel marker developed in this study will be useful for the identification of *L. officinale* and authentication of Cnidii Rhizoma.

## 3. Materials and Methods

### 3.1. Plant Materials

Fresh leaves of *A. polymorpha* (KIOM201501014664) were collected from natural populations in Korea and used for CP genome sequencing. All samples were assigned identification numbers and registered in the Korean Herbarium of Standard Herbal Resources (Index Herbariorum code KIOM) at the Korea Institute of Oriental Medicine (KIOM, Naju, Korea). Plant samples used for CP genome analysis and indel marker validation are listed in Appendix A.

### 3.2. Sequencing and Assembly of the CP Genome of A. polymorpha

DNA was extracted from leaf samples using DNeasy Plant Maxi Kit (Qiagen, Valencia, CA, USA), according to the manufacturer’s instructions. Illumina short-insert paired-end sequencing libraries were constructed and sequenced using the Illumina MiSeq platform (Illumina, San Diego, CA, USA). CP genome sequences were determined from the de novo assembly of low-coverage whole genome sequences. Trimmed paired-end reads (Phred score ≥20) were assembled using the CLC genome assembler ver. 4.06 beta (CLC Inc., Rarhus, Denmark) with default parameters. Principal contigs representing the CP genome were retrieved from the total collection of contigs using Nucmer [55] and aligned with the reference CP genome sequence of *Angelica acutiloba* (KT963036). De novo SOAP gap closer was performed to fill gaps based on the aligned paired-end reads [56].

### 3.3. Annotation and Comparative Analysis

Gene annotation of the CP genome of *A. polymorpha* was performed using GeSeq [57], and annotation results were concatenated using an in-house script pipeline. Protein-coding sequences were manually curated and confirmed using Artemis [58] and checked against the NCBI protein database. Sequences of tRNAs were confirmed using tRNAscan-SE 1.21 [59], and those of the IR regions were confirmed using IR finder and RepEx [60,61]. Circular maps of the *A. polymorpha* CP genome were generated using OGDRAW [62]. The GC content and RSCU values were analyzed using MEGA6 software [63]. Sequences of LSC/IR, IR/SSC, SSC/IR, and IR/LSC junctions were validated using PCR-based sequencing. Primer information and sequence alignment results are listed in Appendix A and Appendix A, respectively. CP genome sequence reads were mapped onto the complete genome using Burrows-Wheel Aligner ver. 0.7.25 [64]. The complete CP genome sequence of *A. polymorpha* was deposited in NCBI under the accession number MH260705. Comparative analysis of the CP genomes of *A. polymorpha* and *L. officinale* was performed using the mVISTA program in Shuffle-LAGAN mode, with the *A. polymorpha* CP genome as a reference [65]. DnaSP version 5.1 was used to calculate nucleotide diversity (*Pi*) between *A. polymorpha* and *L. officinale* CP genomes [66].

### 3.4. Analysis of SSRs and Tandem and Palindromic Repeats in CP Genomes of A. polymorpha and L. officinale

Tandem repeats were at least 20 bp in length, with minimum alignment score and maximum period size set at 50 and 500, respectively. The identity of repeats was set at ≥90%. SSRs were detected using MISA, with minimum repeat number set at 10, 5, 4, 3, 3, and 3 for mono-, di-, tri-, tetra-, penta- and hexanucleotides, respectively [67]. The IR regions were detected using the Inverted Repeats Finder with default parameters. The IR regions were required to be at least 20 bp in length with 90% sequence similarity [68].

### 3.5. Phylogenetic Analysis

A total of 33 CP genomes, including 23 Apiaceae, eight Araliaceae, and two outgroup species (*Adoxa moschatellina* (GenBank accession no. NC_034792) and *Tetradoxa omeiensis* (GenBank accession no. NC_034794)), were used for phylogenetic analysis. Of these, 32 CP genomes were downloaded from the NCBI GenBank database (Appendix A). Alignments of 52 conserved protein-coding genes were used to construct molecular phylogenetic trees with MAFFT [69] and then manually adjusted using Bioedit [70]. The best-fitting nucleotide substitution model was determined using the Akaike Information Criterion (AIC) in JModeltest V2.1.10 [71]. The GTR+I+G model was used in both species. ML analysis was performed using MEGA6 with 1000 bootstrap replicates [59], and BI analysis was performed in MrBayes 3.2.2 using the Markov Chain Monte Carlo (MCMC) method, with two independent runs (four chains each) for one million generations. Phylogenetic trees were sampled every 1000 generations, with the first 25% discarded as burn-in. Phylogenetic trees were determined from 50% majority-rule consensus trees to estimate PP values [72].

### 3.6. Development and Validation of the LYCE Indel

Regions containing indels were selected based on sequence similarities detected with mVISTA. To amplify these regions, primers were designed using Primer-BLAST (NCBI). Indel regions were amplified from 20 ng of genomic DNA in a 20 µL reaction volume containing Solg™ 2X *Taq* PCR Smart Mix 1 (Solgent, Daejeon, Korea) and 10 pmol of each primer (Bioneer, Daejeon, Korea). Amplification was performed on a Pro Flex PCR system (Applied Biosystems, Waltham, MA, USA) under the following conditions: initial denaturation at 95 °C for 2 min, followed by 35 cycles of denaturation at 95 °C for 40 s, annealing at 60 °C for 40 s, and extension at 72 °C for 50 s, and lastly a final extension at 72 °C for 5 min. PCR products were separated on 2% agarose gels at 150 V for 40 min. The specificity of the indel marker and variability in indel regions between *A. polymorpha* and *L. officinale* were verified based on PCR amplification profiles of all 21 samples. All samples were assigned identification numbers, and voucher specimens were deposited in the Korean Herbarium of Standard Herbal Resources (IH code KIOM). In addition, to confirm that PCR product sizes were accurate, each PCR product was isolated using a gel extraction kit (Qiagen, Valencia, CA, USA), subcloned into the pGEM-T Easy vector (Promega, Madison, WI, USA), and sequenced on a DNA sequence analyzer (ABI 3730, Applied Biosystems Inc., Foster City, CA, USA).

## 4. Conclusions

In this study, we sequenced the CP genome of *A. polymorpha* and compared it with that of *L. officinale*. The CP genomes of both species were highly conserved with respect to gene content, gene orientation, and GC content; however, local sequence variations were detected between *A. polymorpha* and *L. officinale*. The most divergent regions between the two CP genomes were found in three non-coding IGS regions (*trnT-psbD*, *ycf4-cemA*, and *ycf2-trnL*) and two genic regions (*ycf1* and *ycf2*). Analysis of nucleotide diversity revealed the highest diversity in the *ycf4-cemA* region. The results of phylogenetic analysis of CP genomes were consistent with those of previous studies. Additionally, we developed a novel indel marker, LYCE, based on sequence variation in the *ycf4-cemA* region to discriminate the herbal medicine *L. officinale* from the adulterant *A. polymorpha*. Thus, analysis of CP genomes is key for species identification, taxonomic classification, and evolutionary analysis of the Apiaceae family members. The LYCE indel marker will be useful for the authentication of Cnidii Rhizoma.

## Figures and Tables

**Figure 1 molecules-24-01038-f001:**
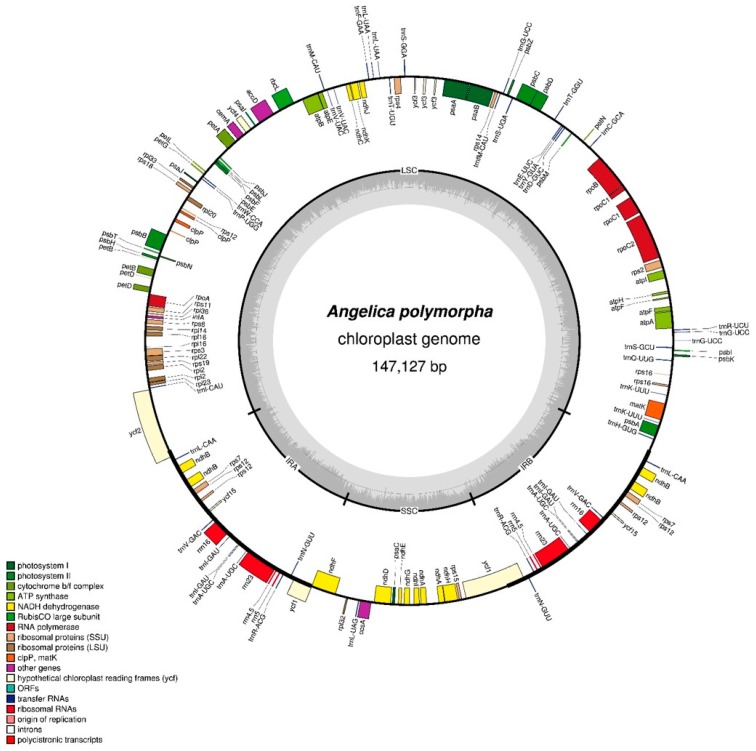
Circular gene map of the CP genome of *A. polymorpha*. Genes drawn inside the circle are transcribed clockwise, and those drawn outside the circle are transcribed counterclockwise. The darker gray inner circle represents the GC content.

**Figure 2 molecules-24-01038-f002:**
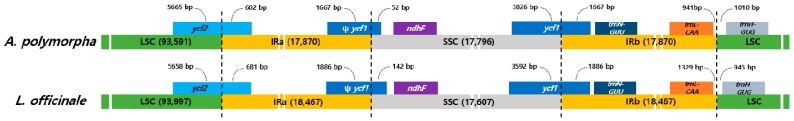
Comparison of CP genome sequences of *A. polymorpha* and *L. officinale* at the junctions of the LSC, IR (IRa and IRb), and SSC regions. ψ: pseudogenes.

**Figure 3 molecules-24-01038-f003:**
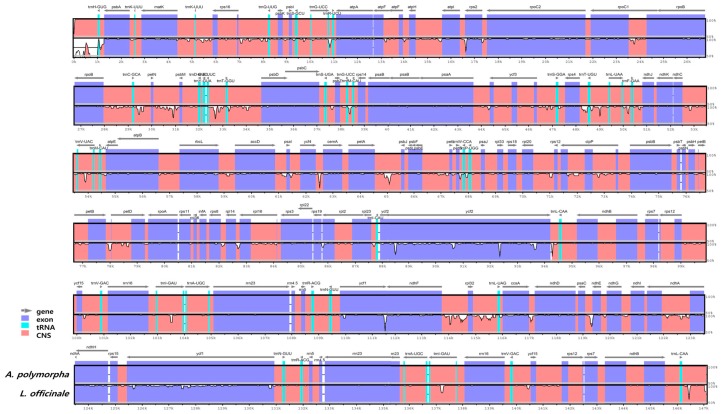
Comparative analysis of the CP genomes of *A. polymorpha* and *L. officinale* using mVISTA. Complete CP genomes of the two species were compared, with the CP genome of *A. polymorpha* used as a reference. Blue block, conserved genes; sky-blue block, tRNA and rRNA genes; red block, conserved non-coding sequences (CNSs); white block, regions polymorphic between *A. polymorpha* and *L. officinale*.

**Figure 4 molecules-24-01038-f004:**
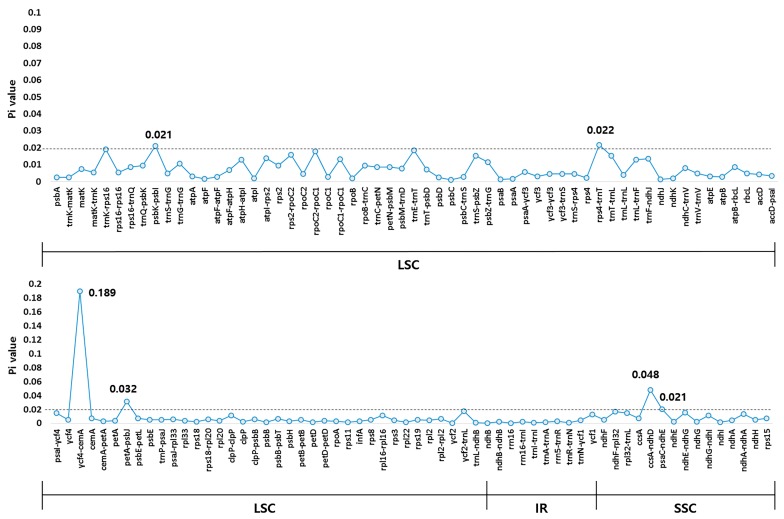
Comparison of nucleotide diversity (*Pi*) between the CP genomes of *A. polymorpha* and *L. officinale*.

**Figure 5 molecules-24-01038-f005:**
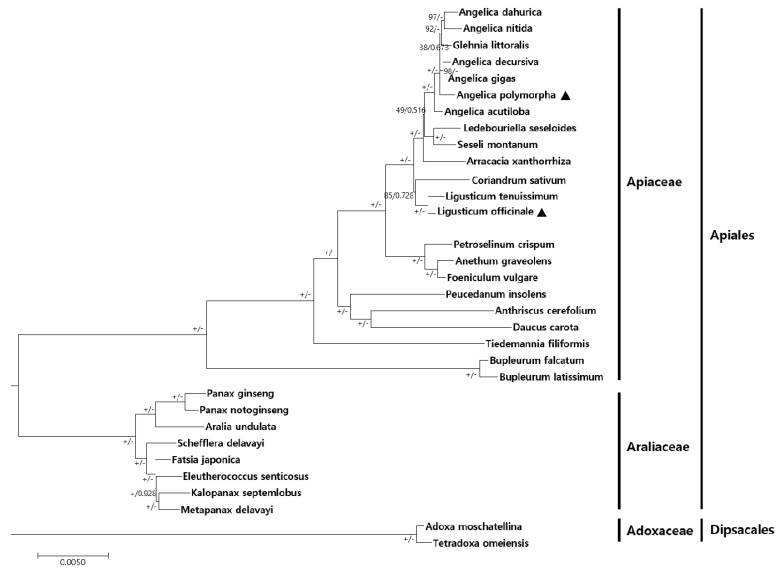
Phylogenetic tree showing the relationship of *A. polymorpha* with 31 species based on 52 protein-coding genes using maximum likelihood (ML) and Bayesian inference (BI) posterior probabilities. The ML topology is indicated with ML bootstrap support values and BI posterior probabilities at each node. The ‘+’ sign indicates ML bootstrap values of 100%, and the ‘–’ sign indicates BI posterior probabilities of 1.0. Black triangles represent the CP genomes of *A. polymorpha* and *L. officinale* examined in this study.

**Figure 6 molecules-24-01038-f006:**
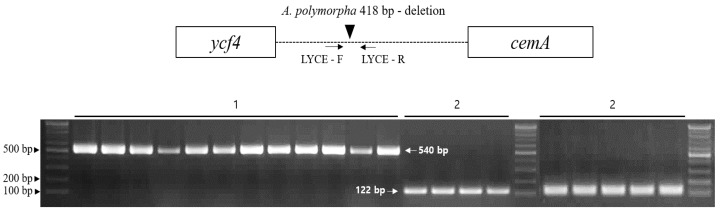
PCR amplification of the LYCE indel marker in 21 *A. polymorpha* and *L. officinale* accessions. 1, *L. officinale*; 2, *A. polymorpha.*

**Table 1 molecules-24-01038-t001:** Characteristics of the CP genomes of *A. polymorpha* and *L. officinale*.

Characteristic ^1^	*A. polymorpha*	*L. officinale* ^2^
Accession number	MH260705	NC039760 [34]
Genome size		
Total CP genome (bp)	147,127	148,518
Large single copy (LSC) region (bp)	93,591	93,977
Inverted repeat (IR) region (bp)	17,870	18,467
Small single copy (SSC) region (bp)	17,796	17,607
Number of unique genes		
Total	113	113
Protein-coding genes	79	79
rRNA genes	4	4
tRNA genes	30	30
GC content (%)		
Total genome	37.5	37.6
LSC region	35.9	36.0
IR regions	45.0	44.8
SSC region	31.0	31.1

^1^ CP: Chloroplast; LSC: Large single copy; IR: Inverted repeat; SSC: Small single copy. ^2^ CP genome of *L. officinale* was downloaded from GenBank.

**Table 2 molecules-24-01038-t002:** Primers used for the development of the indel marker.

Primer Name	Primer Sequence (5′→3′)	Position
LYCE-F	CGC TCA TTC TAG TCA AAG AAG ACG	*ycf4-cemA*
LYCE-R	CGC CAT CCA ATA TTT CTC TCA TGC	

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
