# Peer review of "Sequencing and Comparative Analysis of the Chloroplast Genome of Angelica polymorpha and the Development of a Novel Indel Marker for Species Identification"

_molecules, 2019, doi:10.3390/molecules24061038_

Round 1
Reviewer 1 Report
Review Report
This study presents comprehensive data on the complete chloroplast genome of Angelica polymorpha. Moreover, a novel indel marker was established to discriminate between this species and other herbal medicines, Ligusticum officinale in particular, due to the high similarity of their sliced preparations to the naked eye. The placement of these two species within the Apiaceae family was also achieved by phylogenetic analyses conducted in this study.
Broad comments:
A revision is still necessary to make this manuscript clearer and more concise. Specific comments/suggestions endorse the aspects that, in my opinion, need reformulation.
Specific comments
Abstract
Lines 26, 27: Please indicate full meaning of “ycf1”, “ycf2”, “ycf4-cemA”.
This also applies to all first appearance of abbreviations along the manuscript.
Line 28: “…revealed that A. polymorpha and L. officinale were well clustered…” NOT “…revealed A. polymorpha and L. officinale was well clustered…”
Introduction
Line 39: “…180 Kbp…” NOT “… 180 Kb…”
Line 44: “as well as parasitic plants”. Why this sentence since parasitic plants are angiosperms?
I think you rather mean: “…such as parasitic plants”?
Lines 53 - 54: This sentence needs reformulation/explanation of the term “utility”: “… to understand the utility of herbal medicines in the genus Angelica”
Line 75:” … such as a molecular marker…” NOT “… such as molecular marker…”
Results and Discussion
Line 86: “…showed a quadripartite structure in most land plants …” I do not understand this sentence. Maybe you want to say: “…showed a quadripartite structure like in most land plants …”
Line 116: “Analysis of repeated sequences …” NOT “Repeat analysis…”
Line 117: “… repeated sequences” NOT “…repeat sequences”
Line 155: “…which was the highest” NOT “…which was highest”
Line 168: “The maximum likelihood (ML) tree and Bayesian inference (BI) trees …” NOT “The maximum likelihood (ML) tree and Bayesian inference (BI) …”
Line 170: Confusing sentence. Suggestion: “…. results strongly indicated that show Apiaceae and …”
Line 172: Suggestion: “together…” NOT “within a monophyletic group…”.
Lines 173 - 174: I cannot agree with this sentence. It is true that they appear separated from the other Apiaceae. However, they cluster with a very low PP value (0.728) and a medium BS value (85).
Lines 175 - 177: Please consider the inclusion of these observations above, in line 172. However, contrarily to what was stated, what is shown in the trees is that the genus Angelica is highly supported (PP=1; BS=100).
Line 177: Concerning the clustering of Glehnia littoralis with Angelica, please consider the very low BS and PP values registered. Therefore, the sentence in line 182 should also be reformulated.
Line 208 - 214: The word "accessions" was over-used! All these sentences may be improved.
Materials and Methods
Lines 256 - 257: “…Table S8 and Table S2, respectively.” NOT “…Table S2 and Table S8, respectively.”
Line 295: Please indicate the meaning of “KIOM”
Line 296: Before sub-cloning, no need of purification of gel extracted products?
An additional comment to the methodology indicated in Line 296: Why did you not perform the direct sequence of purified gel bands?
Author Response
Reviewer 1
Open Review
(x) I would not like to sign my review report
( ) I would like to sign my review report
English language and style
( ) Extensive editing of English language and style required
(x) Moderate English changes required
( ) English language and style are fine/minor spell check required
( ) I don't feel qualified to judge about the English language and style
Yes | Can be improved | Must be improved | Not applicable | |
Does the introduction provide sufficient background and include all relevant references? | ( ) | (x) | ( ) | ( ) |
Is the research design appropriate? | (x) | ( ) | ( ) | ( ) |
Are the methods adequately described? | ( ) | (x) | ( ) | ( ) |
Are the results clearly presented? | ( ) | ( ) | (x) | ( ) |
Are the conclusions supported by the results? | (x) | ( ) | ( ) | ( ) |
Comments and Suggestions for Authors
Review Report
This study presents comprehensive data on the complete chloroplast genome of Angelica polymorpha. Moreover, a novel indel marker was established to discriminate between this species and other herbal medicines, Ligusticum officinale in particular, due to the high similarity of their sliced preparations to the naked eye. The placement of these two species within the Apiaceae family was also achieved by phylogenetic analyses conducted in this study.
Broad comments:
A revision is still necessary to make this manuscript clearer and more concise. Specific comments/suggestions endorse the aspects that, in my opinion, need reformulation.
Response: Thank you for your comments. We revised the manuscript according to your suggestions.
Specific comments
Abstract
- Lines 26, 27: Please indicate full meaning of “ycf1”, “ycf2”, “ycf4-cemA”.
This also applies to all first appearance of abbreviations along the manuscript.
Response: we revised manuscript as your suggestion (lines 26-28).
-Line 28: “…revealed that A. polymorpha and L. officinale were well clustered…” NOT “…revealed A. polymorpha and L. officinale was well clustered…”
Response: We revised. Line 29
Introduction
-Line 39: “…180 Kbp…” NOT “… 180 Kb…”
Response: We revised. Line 40
-Line 44: “as well as parasitic plants”. Why this sentence since parasitic plants are angiosperms? I think you rather mean: “…such as parasitic plants”?
Response: We revised. Lines 45-46
-Lines 53 - 54: This sentence needs reformulation/explanation of the term “utility”: “… to understand the utility of herbal medicines in the genus Angelica”
Response: We revised. Lines 54-55
-Line 75:” … such as a molecular marker…” NOT “… such as molecular marker…”
Response: We revised. Line 76
Results and Discussion
-Line 86: “…showed a quadripartite structure in most land plants …” I do not understand this sentence. Maybe you want to say: “…showed a quadripartite structure like in most land plants …”
Response: We revised. Line 87
-Line 116: “Analysis of repeated sequences …” NOT “Repeat analysis…”
Response: We revised. Line 117
-Line 117: “… repeated sequences” NOT “…repeat sequences”
Response: We revised. Line 118
-Line 155: “…which was the highest” NOT “…which was highest”
Response: We revised. Lines 156-157
-Line 168: “The maximum likelihood (ML) tree and Bayesian inference (BI) trees …” NOT “The maximum likelihood (ML) tree and Bayesian inference (BI) …”
Response: We revised. Lines 169-170
-Line 170: Confusing sentence. Suggestion: “…. results strongly indicated that show Apiaceae and …”
Response: We revised as your suggestions. In our phylogenic trees, species within Apiaceae and Arliaceae were completely divided. Thus, we mentioned two families are well clustered. Line 171
-Line 172: Suggestion: “together…” NOT “within a monophyletic group…”.
Response: we revised manuscript as your suggestions (line 173).
-Lines 173 - 174: I cannot agree with this sentence. It is true that they appear separated from the other Apiaceae. However, they cluster with a very low PP value (0.728) and a medium BS value (85).
Response: We revised as your suggestion. Line 173-174
-Lines 175 - 177: Please consider the inclusion of these observations above, in line 172. However, contrarily to what was stated, what is shown in the trees is that the genus Angelica is highly supported (PP=1; BS=100).
Response: We revised manuscript. Lines 176 - 178
-Line 177: Concerning the clustering of Glehnia littoralis with Angelica, please consider the very low BS and PP values registered. Therefore, the sentence in line 182 should also be reformulated.
Response: We removed this sentence in revised manuscript.
-Line 208 - 214: The word "accessions" was over-used! All these sentences may be improved.
Response: We revised. Lines 208-214
Materials and Methods
-Lines 256 - 257: “…Table S8 and Table S2, respectively.” NOT “…Table S2 and Table S8, respectively.”
Response: We revised. Lines 255-256
-Line 295: Please indicate the meaning of “KIOM”
Response: We revised. Lines 293 - 295
-Line 296: 1) Before sub-cloning, no need of purification of gel extracted products? An additional comment to the methodology indicated in Line 296: 2) Why did you not perform the direct sequence of purified gel bands?
Response: We revised as your suggestion. Lines 295 - 296
1) We extracted gel using gel extraction kit (Qiagen). This kit is applied to extraction and purification. Thus, our experiment was adopted purification process from gel extraction kit following the manufacturer’s instructions.
2) To check more accurate band size for amplified PCR products and to avoid errors during PCR amplification and sequencing, and to identify potential chimeric sequences, we performed sub-cloning approach. The inserted DNA fragments were sequenced using the primers SP6 and T7, using an ABI3730 DNA sequence analyzer. Sometimes, direct sequencing indicated highly sequencing errors that flanking start and end region. To check sequencing errors, we adopted sub-cloning method.
Reviewer 2 Report
line 100 - it is not true as ycf1 in IRa is a pseudogene of different size than that in IRb
line 136 - concerns ycf1: it should be described more precisely as these are not the same structures
line 140 - this figure is drawn not in scale, eg. 52bp of A. polymorpha is the same size as 142bp in L. officinale and so on - schould be corrected
line 215-217 - this is repetition of the previous statement from introduction
Please, check every place describing ycf1 for pseudogene correction.
Author Response
Reviewer2
Open Review
(x) I would not like to sign my review report
( ) I would like to sign my review report
English language and style
( ) Extensive editing of English language and style required
( ) Moderate English changes required
( ) English language and style are fine/minor spell check required
(x) I don't feel qualified to judge about the English language and style
Yes | Can be improved | Must be improved | Not applicable | |
Does the introduction provide sufficient background and include all relevant references? | (x) | ( ) | ( ) | ( ) |
Is the research design appropriate? | (x) | ( ) | ( ) | ( ) |
Are the methods adequately described? | (x) | ( ) | ( ) | ( ) |
Are the results clearly presented? | ( ) | (x) | ( ) | ( ) |
Are the conclusions supported by the results? | (x) | ( ) | ( ) | ( ) |
Comments and Suggestions for Authors
-line 100 - it is not true as ycf1 in IRa is a pseudogene of different size than that in IRb
Response: We revised sentence as your suggestions. Lines 100-101
-line 136 - concerns ycf1: it should be described more precisely as these are not the same structures
Response: We revised manuscript. Line 137
-line 140 - this figure is drawn not in scale, eg. 52bp of A. polymorpha is the same size as 142bp in L. officinale and so on - schould be corrected
Response: We revised Figure 2, and marked ycf1 and pseudogenes, respectively. Lines 140 -142
-line 215-217 - this is repetition of the previous statement from introduction
Response: We removed repetitive sentence in the part of Results and Discussion.
-Please, check every place describing ycf1 for pseudogene correction.
Response: We confirmed and revised for ycf1 in revised manuscript.